EMBO
Molecular Medicine

# Chimeric *Pneumoviridae* fusion proteins as immunogens to induce cross-neutralizing antibody responses

Eduardo Olmedillas[1,2], Olga Cano[1,2], Isidoro Martínez[1], Daniel Luque[1], María C Terrón[1], Jason S McLellan[3], José A Melero[1,2,*] [iD] & Vicente Más[1,2]

## Abstract

Human respiratory syncytial virus (hRSV) and human metapneumovirus (hMPV), two members of the *Pneumoviridae* family, account for the majority of severe lower respiratory tract infections worldwide in very young children. They are also a frequent cause of morbidity and mortality in the elderly and immunocompromised adults. High levels of neutralizing antibodies, mostly directed against the viral fusion (F) glycoprotein, correlate with protection against either hRSV or hMPV. However, no cross-neutralization is observed in polyclonal antibody responses raised after virus infection or immunization with purified F proteins. Based on crystal structures of hRSV F and hMPV F, we designed chimeric F proteins in which certain residues of well-characterized antigenic sites were swapped between the two antigens. The antigenic changes were monitored by ELISA with virus-specific monoclonal antibodies. Inoculation of mice with these chimeras induced polyclonal cross-neutralizing antibody responses, and mice were protected against challenge with the virus used for grafting of the heterologous antigenic site. These results provide a proof of principle for chimeric fusion proteins as single immunogens that can induce cross-neutralizing antibody and protective responses against more than one human pneumovirus.

**Keywords** neutralizing antibodies; *Pneumoviridae*; structure-based design; universal vaccines

**Subject Categories** Immunology; Microbiology, Virology & Host Pathogen Interaction

## Introduction

Human respiratory syncytial virus (hRSV) is the leading cause worldwide of severe acute lower respiratory tract infections (ALRI, mainly bronchiolitis and pneumonia) in infants and very young children. The estimated new cases of ALRI each year owing to hRSV infections are more than 33 million in children younger than 5 years, with 10% of these resulting in hospital admissions. hRSV infections also cause 60,000–120,000 deaths each year in young children, with 90% occurring in developing countries (Nair *et al*, 2010; Shi *et al*, 2017). Human metapneumovirus (hMPV) is also a virus of clinical significance (van den Hoogen *et al*, 2001), second only to hRSV as a cause of ALRI in young children (Edwards *et al*, 2013; Schuster & Williams, 2013). Both hRSV and hMPV also have a clinical impact in the elderly and in adults with cardiopulmonary disease or impaired immune systems (Hall *et al*, 1986; Dowell *et al*, 1996; Falsey *et al*, 2005; Panda *et al*, 2014). hRSV and hMPV have been recently reclassified in the *orthopneumovirus* and *metapneumovirus* genera, respectively, of the newly created *Pneumoviridae* family, which was detached from the original *Paramyxoviridae* family (Afonso *et al*, 2016). Hence, hRSV and hMPV share clinical and biological features, although they also have important differences, for example, in the number of genes or in the proteolytic processing of their respective fusion (F) glycoproteins (van den Hoogen *et al*, 2002).

Both hRSV and hMPV genomes encode three glycoproteins (SH, G, and F) that are inserted in the viral membrane. SH is a small hydrophobic protein whose function is still unclear and is incorporated in low amounts into the virus particle (Bao *et al*, 2008; Gan *et al*, 2012). G is a heavily glycosylated and highly variable type II glycoprotein that resembles mucins and serves as the viral attachment protein (Levine *et al*, 1987; Thammawat *et al*, 2008). Finally, F is a type I glycoprotein that fuses the viral and cell membranes at the initial stages of an infectious cycle and is the main target of neutralizing antibodies (for a recent review Melero & Mas, 2015).

The F protein is synthesized as a F0 precursor that requires proteolytic processing to become functional. While hMPV F is cleaved only once by trypsinlike proteases outside the cell (Shirogane *et al*, 2008), hRSV F is cleaved twice inside the cell at two polybasic sites recognized by furinlike proteases (Gonzalez-Reyes *et al*, 2001; Zimmer *et al*, 2001; Begona Ruiz-Arguello *et al*, 2002).

1   Centro Nacional de Microbiología, Instituto de Salud Carlos III, Madrid, Spain
2   CIBER de Enfermedades Respiratorias, Instituto de Salud Carlos III, Madrid, Spain
3   Department of Biochemistry and Cell Biology, Geisel School of Medicine at Dartmouth, Hanover, NH, USA
    *Corresponding author. Tel: +34 9182 23908; E-mail: jmelero@isciii.es

Both hRSV F and hMPV F are trimers of disulfide-linked heterodimers, and these F proteins initially fold into a metastable prefusion conformation. During membrane fusion, the F glycoprotein refolds through a series of unstable intermediates into a highly stable postfusion conformation that shares some neutralizing epitopes with the prefusion conformation (Lamb *et al*, 2006; McLellan *et al*, 2013b).

There are no available vaccines for either hRSV or hMPV despite being greatly needed. A wealth of data support the conclusion that neutralizing antibodies are the major players in protection against hRSV and hMPV infections (for a recent review Melero & Mas, 2015). The majority of these antibodies are directed against the F glycoprotein (Walsh & Hruska, 1983), and it was recently shown that most of the hRSV-neutralizing activity present in human sera was due to antibodies specific for the metastable prefusion F conformation (Magro *et al*, 2012; Ngwuta *et al*, 2015).

The search for new hRSV vaccines has been reinvigorated by recent advances in structure-based design of soluble hRSV F proteins folded in either the prefusion (McLellan *et al*, 2013a; Krarup *et al*, 2015) or postfusion conformations (McLellan *et al*, 2011b; Swanson *et al*, 2011). Stabilized soluble forms of prefusion hRSV F were shown to induce higher levels of neutralizing antibodies than soluble postfusion hRSV F in mice, cotton rats, and nonhuman primates (McLellan *et al*, 2013a; Krarup *et al*, 2015; Palomo *et al*, 2016). However, postfusion F, which has the advantage of being highly stable, can induce sizeable levels of neutralizing antibodies and afford protection against hRSV because it shares certain epitopes with prefusion F (Swanson *et al*, 2011). Recently, the structure of a soluble form of postfusion hMPV F was determined, revealing extensive similarity with postfusion hRSV F despite having only ~38% sequence identity (Mas *et al*, 2016). In addition, the purified postfusion hMPV F protein was able to elicit high titers of neutralizing antibodies in mice (Mas *et al*, 2016).

A few monoclonal antibodies capable of neutralizing both hRSV and hMPV have been reported (Corti *et al*, 2013; Schuster *et al*, 2014; Mas *et al*, 2016); however, no significant cross-neutralization was detected in polyclonal antibody responses elicited by soluble postfusion forms of either hRSV F or hMPV F (Mas *et al*, 2016). Based on knowledge gained about the antigenic structure of these two proteins, we engineered a series of chimeric proteins in which certain epitopes of hRSV F were grafted onto hMPV F and vice versa. When mice were inoculated with the purified chimeras, they elicited antibodies that neutralized both hRSV and hMPV. Furthermore, in cases that were tested, immunization of mice with the chimeric proteins afforded protection against a challenge with the virus used for grafting of the heterologous antigenic site. Passive transfer of the mouse sera also reduced significantly lung virus titer after challenge, demonstrating the prominent role of antibodies in the protection provided by the chimeric F proteins.

# Results

The plasmids encoding each of the chimeric proteins shown in Fig EV1 were generated and tested for transient expression of the matching F proteins in CV-1 cells, as described in Materials and Methods. Some chimeric proteins were expressed at very low or undetectable levels and were not further analyzed. Chimeric proteins that reached expression levels above 10% of the wild-type

protein were incorporated into vaccinia virus recombinants for expression and further antigenic and immunogenic characterization as described in subsequent sections.

## Expression and characterization of postfusion hMPV F with antigenic site II from hRSV F

Based on the similarity of hMPV F and hRSV F antigenic site II structures (Mas *et al*, 2016), it was expected that certain amino acids could be exchanged between the two proteins without disrupting the overall local folding (Mas *et al*, 2016). Indeed, the chimeric proteins F-414, F-415, and F-416 (Fig EV1), with increasing numbers of hRSV F residues replacing the equivalent postfusion hMPV F residues, were expressed at high level and therefore initially analyzed. Since the expression level of F-414 turned out to be the same as that of F-415, the former protein was not further considered.

Much of the C-terminal helix of hMPV F site II in F-415 was made hRSV-like by swapping seven amino acids unique to each protein (Fig 1A and B). Additionally, the chimeric F-416 protein incorporated six further amino acid changes in the hMPV F backbone to reproduce almost entirely the amino acid sequence of hRSV F site II (Fig 1A and B). Both F-415 and F-416 were readily purified to homogeneity by $Ni^{2+}$ chromatography followed by gel filtration and showed the characteristic cone shape of postfusion hMPV F (Mas *et al*, 2016) when examined by negative stain EM (Fig EV2).

When tested for binding to site-specific mAbs, F-415 had lost reactivity with the hMPV F site II mAbs MF9, MF12, MF14, and MF15 but retained reactivity with the hMPV F site IV-specific mAb MF16; in addition, it gained full and partial reactivity with the hRSV F site II mAbs motavizumab (Mz) and 47F, respectively (Fig 1C). F-416 also lost reactivity with hMPV F site II-specific mAbs but gained full reactivity not only with Mz but also with 47F (Fig 1C).

To have a better estimate of the affinities of Mz and 47F for the chimeric proteins, the binding of their respective Fab fragments to wild-type postfusion hRSV F and the chimeric F proteins was assessed by surface plasmon resonance (SPR) (Fig 2). Mz Fab bound to the F-415 and F-416 chimeras with a $k_{on}$ similar to that observed for the wild-type postfusion hRSV F but showed a much faster $k_{off}$ for F-415 than for the wild-type and F-416 proteins. Consequently, Mz affinity for the F-416 chimera was similar to that for postfusion hRSV F but was markedly decreased for F-415 (see $K_D$ values).

No binding of 47F Fab to F-415 was detectable (Fig 2), and although binding of 47F Fab to F-416 was quantifiable—with a slightly faster $k_{on}$ compared with wild-type postfusion hRSV F—a much faster $k_{off}$ reduced its affinity for F-416 by 10-fold compared to postfusion hRSV F. Therefore, although site II of F-415 and particularly F-416 protein resembled the antigenic properties of this site in hRSV F, there were some remaining distinguishable differences between wild-type and chimeric proteins.

We noted that although ELISA binding of Mz and 47F to the chimeric proteins followed the same trend as that of their Fab fragments in SPR, significant differences could be perceived with the two methods. Thus, whereas binding of 47F Fab to F-415 was negligible by SPR (Fig 2), substantial binding of mAb 47F to F-415 was observed in the ELISA (Fig 1C). Similarly, although binding of mAb Mz to F-415 was comparable to postfusion hRSV F in the ELISA, a

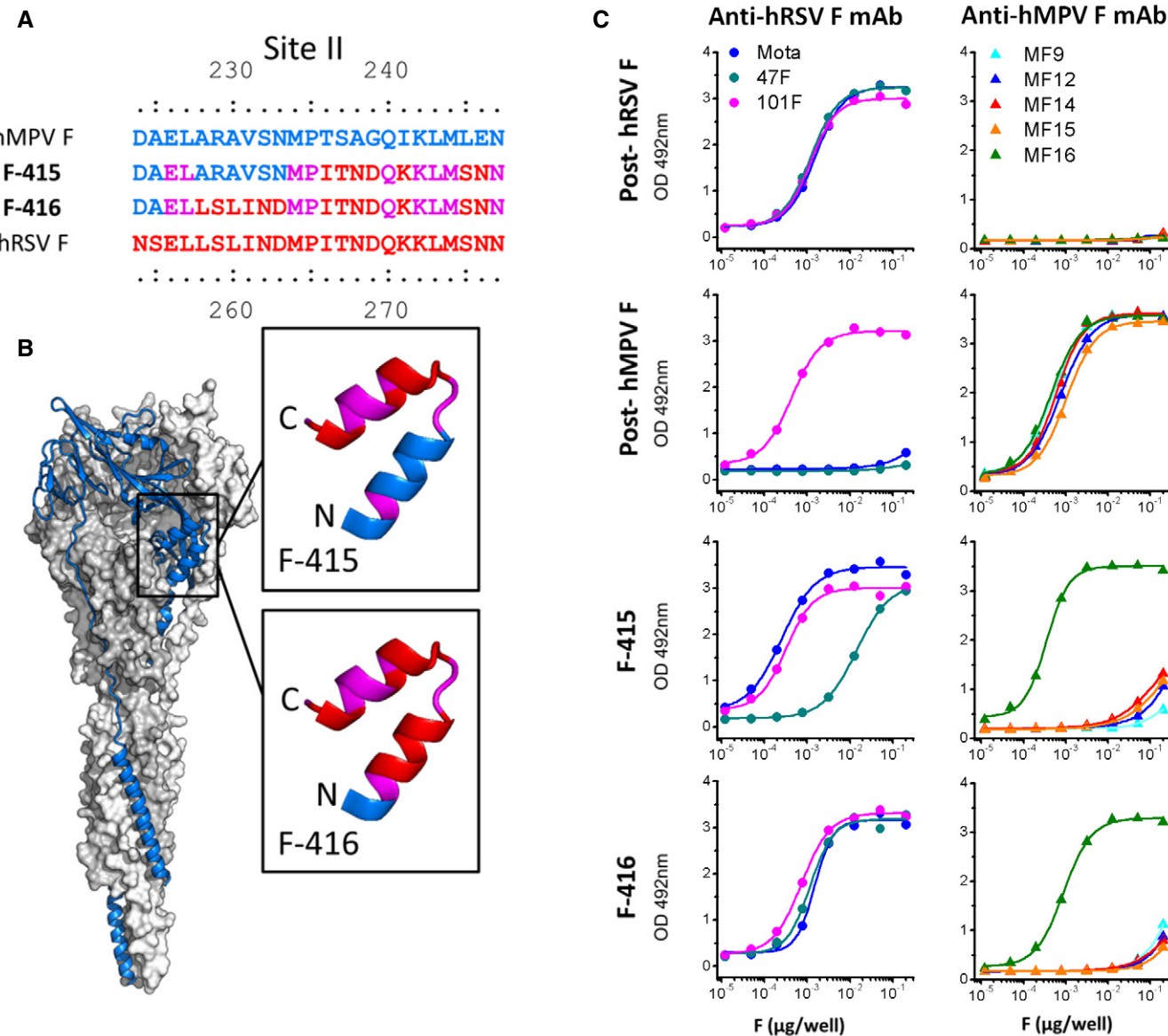

**Figure 1. Antigenic characterization of postfusion hMPV F chimeras with amino acids from hRSV F antigenic site II.**

A  Partial amino acid sequences of the hMPV F (blue) and hRSV F (red) antigenic site II (Toiron *et al*, 1996; Lopez *et al*, 1998; McLellan *et al*, 2011b). Partial sequences of the F-415 and F-416 chimeras are color coded to indicate the origin of their antigenic site II residues. Amino acids shared by hMPV F and hRSV F are colored purple. The remaining sequences of F-415 and F-416 are derived from hMPV F.

B  Surface representation of the hMPV F structure in the postfusion conformation (Mas *et al*, 2016). One of the protomers is shown as a blue ribbon. Antigenic site II is magnified and colored as in (A) for the F-415 and F-416 chimeras.

C  ELISA binding results of mAbs specific for hRSV F (left panels) or hMPV F (right panels) with the proteins indicated on the left. mAbs specific for hRSV F bind epitopes of antigenic site II (Mota and 47F) or antigenic site IV (101F). mAs specific for hMPV F bind epitopes of antigenic site II (MF9, MF12, MF14, and MF15) or antigenic site IV (MF16).

substantial reduction in Mz Fab affinity for this chimeric protein was observed by SPR. These differences are likely due to the higher density of proteins bound to the ELISA plates than to the SPR chips and the bivalent nature of antibodies compared with the monovalent Fabs.

To assess the immunogenicity of the F-415 and F-416 chimeras, groups of five mice were inoculated twice (4 weeks apart) i.m. with 10 μg/dose of each purified protein adjuvanted with CpG. The same inoculation regimen was used for postfusion hRSV F and hMPV F, as controls. Three weeks after the last immunization, mice were bled and their sera tested in ELISA for antibody binding to the wild-type postfusion forms of either hRSV F or hMPV F (Fig 3A). As reported, the wild-type postfusion F proteins induced high levels of antibodies that bound to the homologous protein but failed to bind (or bound very poorly in the case of serum from mice inoculated with hMPV F and tested against hRSV F) to the heterologous protein

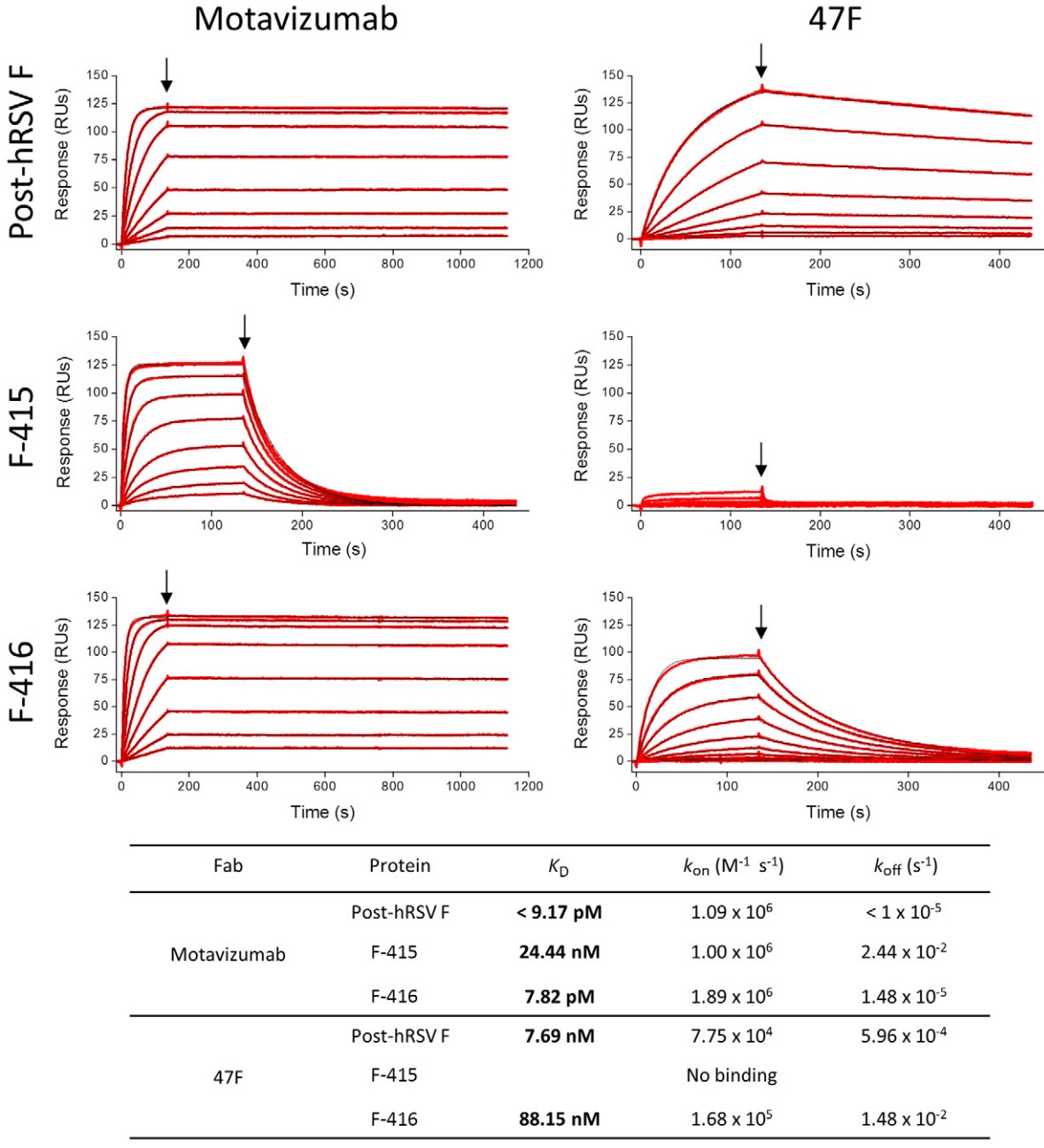

| Fab | Protein | $K_D$ | $k_{on}$ (M$^{-1}$ s$^{-1}$) | $k_{off}$ (s$^{-1}$) |
|---|---|---|---|---|
| Motavizumab | Post-hRSV F | **< 9.17 pM** | 1.09 x 10$^6$ | < 1 x 10$^{-5}$ |
| | F-415 | **24.44 nM** | 1.00 x 10$^6$ | 2.44 x 10$^{-2}$ |
| | F-416 | **7.82 pM** | 1.89 x 10$^6$ | 1.48 x 10$^{-5}$ |
| 47F | Post-hRSV F | **7.69 nM** | 7.75 x 10$^4$ | 5.96 x 10$^{-4}$ |
| | F-415 | | No binding | |
| | F-416 | **88.15 nM** | 1.68 x 10$^5$ | 1.48 x 10$^{-2}$ |

Figure 2. **Binding of Fabs to wild-type and chimeric proteins measured by surface plasmon resonance.**

Binding of Mz and 47F Fabs to the immobilized proteins indicated on the left. Sensorgrams of eight different concentrations are shown in each panel. The calculated binding parameters are shown in the table. Red lines indicate the best fit of the data to a 1:1 binding model. RUs, response units.

(Mas *et al*, 2016). In contrast, the antibodies induced by the F-415 chimera showed substantial binding not only to hMPV F but also to the hRSV postfusion F protein (Fig 3A). The level of antibody binding to postfusion hRSV F was even higher with sera from mice inoculated with F-416 and significantly higher ($P < 0.001$) than sera from mice inoculated with wild-type hMPV F.

To assess whether the antibody binding to hRSV F observed with sera from mice inoculated with F-415 or F-416 was reflected in neutralizing activity against this virus, the mouse sera were tested in a microneutralization assay (Fig 3B). Again, the sera of mice inoculated with the wild-type hMPV F or hRSV F protein had substantial neutralization titers (IC$_{50}$) only against the homologous virus. In contrast, the sera of mice inoculated with either F-415 or F-416 had sizeable neutralization titers against hRSV, in addition to neutralizing hMPV. In summary, the antigenic characteristics of the F-415 and F-416 chimeras, reflected in their reactivity with hRSV F

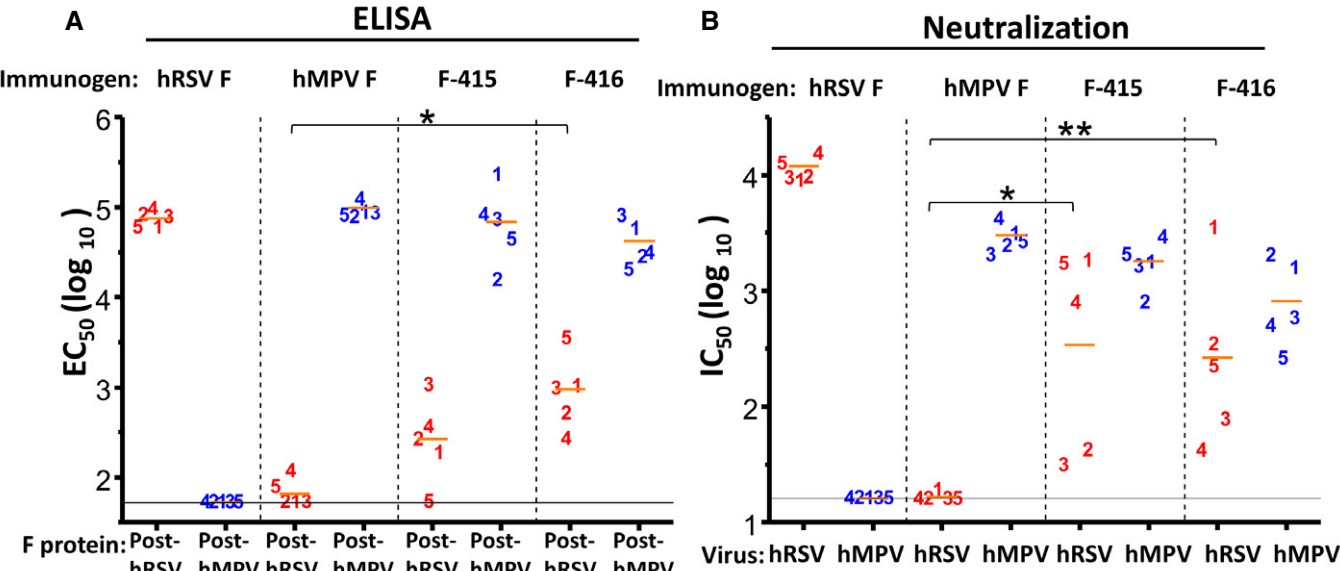

**Figure 3. Antibody responses of mice inoculated twice with either wild-type proteins or chimeric proteins with a postfusion hMPV F backbone.**

Groups of BALB/c female mice ($n$ = 5) were inoculated twice (4 weeks apart) i.m. with 10 µg/dose of the proteins indicated at the top of each panel, folded in their respective postfusion conformations. One week after the last dose, mice were sacrificed and their blood collected.

A    Each mouse serum (identified by a number) was tested in ELISA for antibody binding to the postfusion F proteins indicated at the bottom. The results represented in the $y$-axis for each individual mouse serum correspond to the inverse dilution that gave 50% of maximal binding, expressed in $\log_{10}$ units (EC$_{50}$ ($\log_{10}$)). *$P$ = 0.00045.

B    Mouse sera were also tested in microneutralization assays with hRSV and hMPV, as indicated at the bottom. The results represented in the $y$-axis correspond to the inverse dilution that inhibited 50% of the viral infectivity (IC$_{50}$ ($\log_{10}$)). $P$-values: *$P$ = 0.01001 and **$P$ = 0.00271.

Data information: Each number represents an individual mouse. Short horizontal bars indicate mean values for each group. $P$-values were calculated as indicated under "Statistical Analysis". Only relevant $P$-values are shown. Differences were considered significant when $P$ < 0.05. Long horizontal lines indicate detection limits.

site II-specific mAbs (Fig 1C), correlated with their capacity to induce murine antibodies that bound hRSV F and neutralized hRSV infectivity, which was particularly prominent in the case of F-416. We note, however, that in some sera no quantitative correlation was found between binding and neutralization titers; for example, serum from mouse #5 inoculated with F-415 had low levels of hRSV F-binding antibodies (Fig 3A) but neutralized hRSV very efficiently (Fig 3B).

To evaluate whether induction of hRSV-neutralizing antibodies by the chimeric proteins could be correlated with protection against a challenge with this virus, groups of eight mice were immunized three times, 4 weeks apart, with 20 µg of either F-415 or F-416 adjuvanted with CpG. One week after the last injection, mice were challenged intranasally with hRSV (A2 strain), and 5 days later, the amount of virus in lung extracts was quantified in a plaque assay. In parallel, three groups of five mice were inoculated with either CpG or wild-type postfusion hRSV F or hMPV F, as controls.

The results again demonstrated a significant increase in hRSV-neutralizing antibodies in the sera of mice inoculated with F-415 and F-416, just before the challenge, compared with CpG-only and postfusion hMPV F controls (Fig 4A). Although there was some spread of the hRSV-neutralizing titers of individual mice, those inoculated with F-416 had on average higher neutralization titers than those inoculated with F-415. However, in both groups some individual mouse titers reached values close to those of mice inoculated with postfusion hRSV F.

Reduction in hRSV replication in the lungs of inoculated mice clearly correlated with the noted induction of antibodies that neutralized hRSV infectivity (Fig 4B). Whereas hRSV reached high titers in the lungs of mice inoculated with either CpG or hMPV F, there was no detectable virus in the lungs of mice inoculated with postfusion hRSV F. Two of the mice inoculated with the F-415 chimera (#6 and #7) had no detectable virus in their lungs. These two mice also had the higher titers of hRSV-neutralizing antibodies in their sera (compare Fig 4A and B). As a group, mice inoculated with F-415 protein had a substantial reduction in lung virus titers compared with CpG and hMPV F controls. Remarkably, the lungs of all mice inoculated with F-416 were free of detectable virus, demonstrating the efficacy of F-416 vaccination for protection against hRSV infection.

To further substantiate the relevance of antibodies in protection of mice inoculated with F-415 or F-416 chimeras, sera from each group of mice of Fig 4B were pooled together and passively transferred i.p. to new mice that were challenged the following day with the same amount of virus as in Fig 4. Five days after challenge, the amount of virus in lung extracts was quantified in a plaque assay. The results of Fig EV3 demonstrate that virus replicated to high titers in mice that received sera from those previously inoculated with either CpG or hMPV F. In contrast, no virus was detected in the lungs of mice that received sera from mice previously inoculated with hRSV F. Most importantly, virus titers were significantly reduced in mice that received sera from those previously inoculated

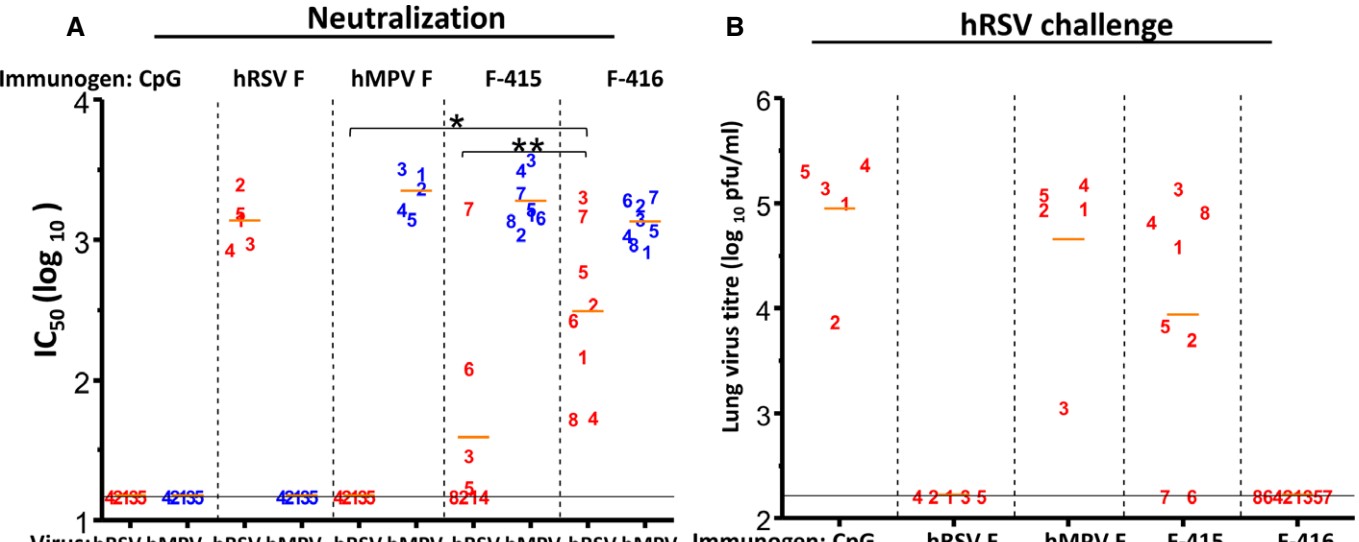

**Figure 4. Protection against hRSV challenge with hMPV F backbone chimeras.**

Groups of BALB/c mice (*n* = 5) were inoculated three times, 4 weeks apart, with 20 µg of the postfusion F proteins indicated above each panel. One group (*n* = 5) was inoculated only with the CpG adjuvant, as indicated. Two other groups of mice (*n* = 8) were similarly inoculated with either F-415 or F-416, as indicated at top of the panels.

A  Mouse sera, collected just before challenging with hRSV (A2 strain), were used in microneutralization assays against the viruses indicated at bottom. *\*P* = 0.00085 and *\*\*P* = 0.01986.

B  Five days after virus inoculation, mice were sacrificed and the amount of virus in lung extracts was measured in a plaque assay. Mean virus titer for each group is indicated by short horizontal bar.

Data information: Each number represents an individual mouse. Short horizontal bars indicate mean values for each group. *P*-values were calculated as indicated under "Statistical Analysis". Only relevant *P*-values are shown. Differences were considered significant when *P* < 0.05. Long horizontal lines indicate detection limits.

with F-415 and particularly with F-416 in which about ten times less virus was found in their lungs than in those from the CpG- and hMPV-negative controls (Fig EV3). It is worth noting though that protection afforded passively by sera from mice previously inoculated with F-416 did not reach the level of protection provided by sera from mice previously inoculated with hRSV F.

**Expression and characterization of prefusion hRSV F with antigenic site IV from hMPV F**

It has been recently demonstrated that most of the neutralizing antibodies in human sera are directed against the prefusion form of hRSV F (Magro *et al*, 2012; Ngwuta *et al*, 2015) and that prefusion-specific site Ø mAbs are the most potent neutralizing antibodies (McLellan *et al*, 2013b; McLellan, 2015). For this reason, and to test the possibility of extending the chimera approach to other conformations and antigenic sites, the chimeric proteins F-410, F-411, and F-412 of Fig EV1 were produced. These proteins were derived from soluble hRSV F stabilized in the prefusion conformation as previously reported (McLellan *et al*, 2013a). Although F-412 was expressed at a moderate level, it failed to react with hMPV F site IV mAbs, and thus, it was discontinued from the study.

In the F-410 protein, five amino acids of antigenic site IV were swapped with the corresponding residues of hMPV F (Fig 5A and B). Six further amino acid changes were introduced in the F-411 chimera to make it more hMPV-like. Both F-410 and F-411 were purified to homogeneity and had the characteristic globular shape of prefusion hRSV F, as seen by EM (Fig EV4).

The antigenic properties of F-410 and F-411 proteins were assessed by ELISA with mAbs specific for the hRSV F and hMPV F proteins. F-410 showed reduced reactivity with mAb 107F (specific for hRSV F site IV) and gained some reactivity with mAb MF16 (specific for site IV of hMPV F) (Fig 5C). Similarly, F-411 showed a complete loss of mAb 107F binding but gained increased reactivity with mAb MF16 and some reactivity, although much reduced, with MF20. mAb 101F, which cross-reacts with both hRSV F and hMPV F site IV (Mas *et al*, 2016), reacted with all wild-type and chimeric proteins (Fig 5C).

To demonstrate further that F-410 and F-411 retained the prefusion characteristics of their hRSV F backbone, their reactivity with conformation-specific antibodies was determined (Fig EV5). Both F-410 and F-411 retained reactivity with three mAbs (D25, AM22, and 5C4) that recognize epitopes in the prefusion-specific site Ø of hRSV F (McLellan *et al*, 2013b). The two chimeric proteins also reacted with two mAbs (14402 and 11569) specific for the recently identified prefusion hRSV F antigenic site V (Gilman *et al*, 2016). In contrast, F-410 and F-411 failed to react with three mAbs (114F, 116F, and 117F) that recognize the six-helix-bundle motif characteristic of postfusion hRSV F (Rodriguez *et al*, 2015).

F-410 and F-411 were then used to immunize groups of five mice (two doses of 10 µg mixed with CpG) to assess their capacity to induce hMPV-specific antibodies. The ELISA binding results revealed essentially no cross-reactivity with the sera of mice inoculated with the prefusion hRSV F or the postfusion hMPV F. However, a substantial increase in hMPV F binding was noted with

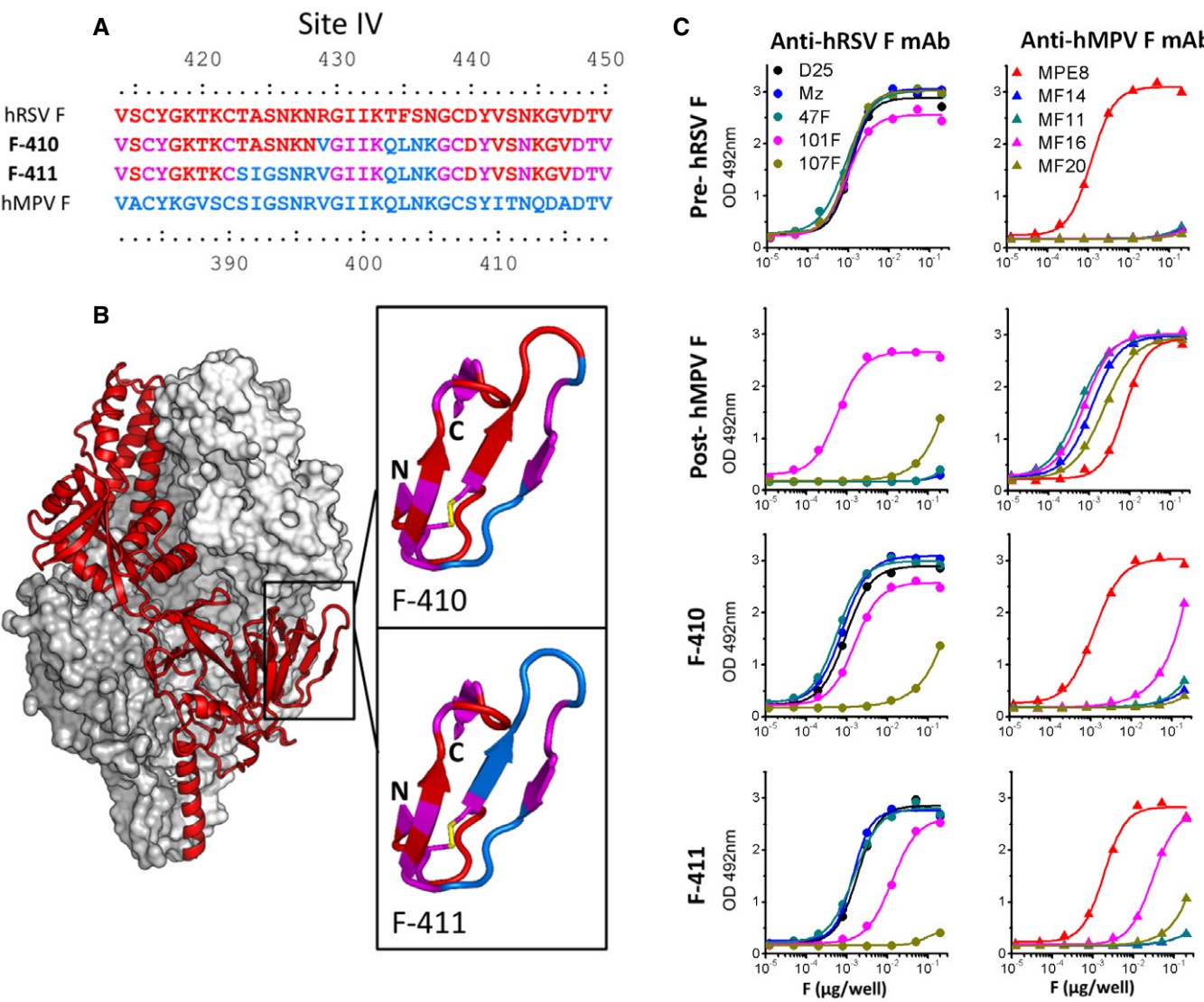

**Figure 5.  Antigenic characterization of prefusion hRSV F chimeras with amino acids from hMPV F antigenic site IV.**

A   Partial amino acid sequences of the hRSV F (red) and hMPV F (blue) antigenic site IV (Lopez *et al*, 1998; Wu *et al*, 2007; McLellan *et al*, 2010). Partial sequences of the F-410 and F-411 chimeras, color coded to indicate the origin of their antigenic site IV residues. Amino acids shared by hMPV F and hRSV F are shown in purple. The remaining sequences of F-410 and F-411 are derived from prefusion hRSV F.

B   Surface representation of the hRSV F structure stabilized in the prefusion conformation, as reported (McLellan *et al*, 2013a). One of the protomers is shown as a red ribbon. Antigenic site IV is magnified and colored as in (A) for the F-410 and F-411 chimeras. Yellow sticks represent a disulfide bridge between Cys416 and Cys422.

C   ELISA binding results of mAbs specific for hRSV F (left panels) or hMPV F (right panels) with the proteins indicated on the left. mAbs specific for hRSV F bind to epitopes of antigenic site Ø (D25), antigenic site II (Mz and 47F), or antigenic site IV (101F and 107F). mAs specific for hMPV F bind to epitopes of antigenic site II (MF14), antigenic site IV (MF11, MF16, and MF29), or an epitope shared by hRSV F and hMPV F (MPE8). 101F epitope is also shared by hRSV F and hMPV F.

sera from mice inoculated with F-410 and particularly F-411, in comparison with sera from mice inoculated with prefusion hRSV F (Fig 6A).

A substantial increase in hMPV neutralization was also noted in the sera of mice inoculated with F-410 and especially F-411, compared with the hRSV F control (Fig 6B). We noted that immunization of mice with either prefusion hRSV F or postfusion hMPV F did not induce significant cross-neutralization, as reported (Mas *et al*, 2016), despite the reports of a few cross-neutralizing mAbs, like 101F (Mas *et al*, 2016) or 54G10 (Schuster *et al*, 2014).

# Discussion

The concept of "universal" vaccines (Nabel & Fauci, 2010) has recently attracted much attention. In some cases, such as influenza virus (Kanekiyo *et al*, 2013; Krammer & Palese, 2014; Yassine *et al*, 2015) or human immunodeficiency virus (HIV) (Kwong *et al*, 2013), a universal vaccine is sought to overcome strain variability so that a single immunogen designed to reorient the antibody response toward relatively conserved antigenic regions would protect against infections with highly divergent antigenic variants. In other cases,

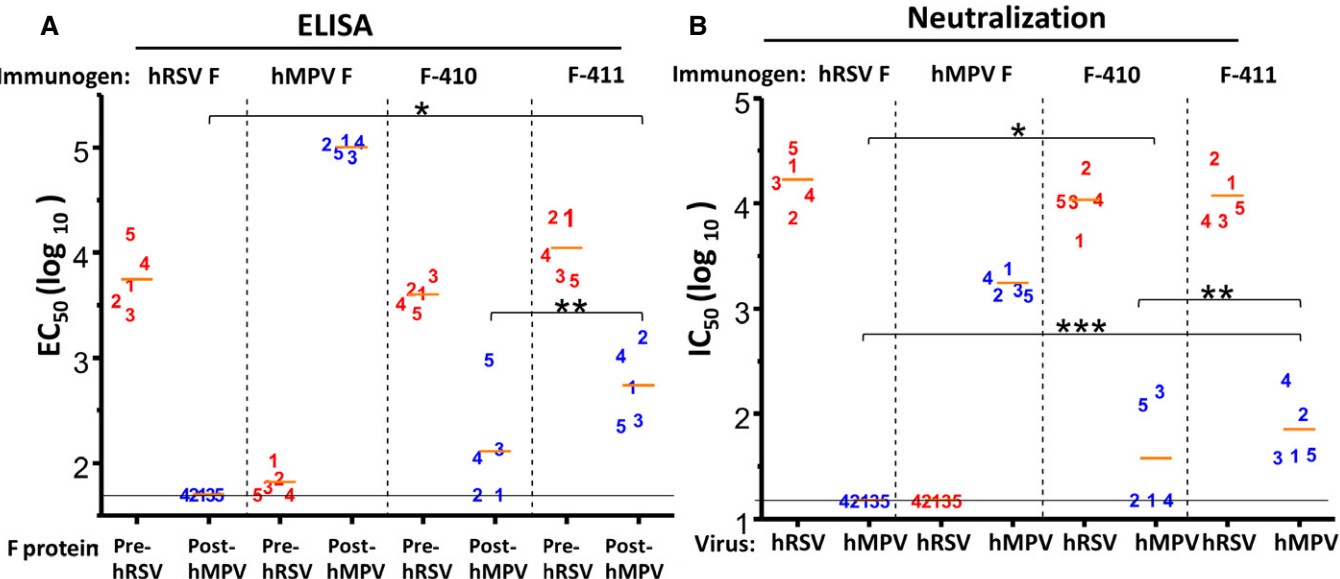

**Figure 6.  Antibody responses of mice inoculated twice with either wild-type or chimeric proteins based on prefusion hRSV F.**

Groups of BALB/c female mice (*n* = 5) were inoculated twice with 10 μg of the proteins indicated at the top of each panel, folded in either the prefusion conformation for hRSV F, F-410, and F-411 or in the postfusion conformation for hMPV F. One week after the last dose, mice were sacrificed and their blood collected.

A    Each mouse serum was tested in ELISA for antibody binding to the F proteins indicated at the bottom of each panel. *P = 0.00046 and **P = 0.02459.

B    Mouse sera were also tested in microneutralization assays with hRSV and hMPV, indicated at the bottom. *P = 0.04138, **P = 0.04831 and ***P = 0.00339.

Data information: Each number represents an individual mouse. Short horizontal bars indicate mean values for each group. *P*-values were calculated as indicated under "Statistical Analysis". Only relevant *P*-values are shown. Differences were considered significant when *P* < 0.05. Long horizontal lines indicate detection limits.

the aim is to develop vaccines that will cross-protect against related viruses from the same family, such as the *Flaviviridae* Zika and dengue viruses (Barba-Spaeth *et al*, 2016). Beyond viruses, universal vaccines are also being designed for *Neisseria meningitidis* with chimeric variants of the factor H binding protein that incorporate epitopes of highly divergent bacterial strains (Scarselli *et al*, 2011).

In this study, we have produced chimeric proteins of hRSV F and hMPV F. Together, these two viruses represent the major global burden of ALRI in very young children. Since palivizumab (Pz), a humanized neutralizing murine mAb (The IMpact-RSV Study Group 1998) currently used for prophylactic treatment of infants, recognizes an epitope of hRSV F antigenic site II, this site has received much attention in the hRSV vaccine field. The structure of hRSV F site II is a helix–turn–helix motif (Toiron *et al*, 1996) conserved in the prefusion and postfusion conformations of the F protein. Thus, hRSV F site II has been grafted onto different protein scaffolds to focus the antibody responses elicited against those unnatural antigens to a site in hRSV F, which is recognized by neutralizing antibodies (McLellan *et al*, 2011a; Correia *et al*, 2014; Luo *et al*, 2015). The results obtained so far have had limited success, since the antibodies induced in mice were not neutralizing despite being able to bind hRSV F. Only in one case, in which site II was grafted onto an immunoglobulin scaffold, did the resulting immunogen induce antibodies in mice that neutralized hRSV *in vitro*, although with relatively low efficiency (Luo *et al*, 2015). In another case, the site II scaffold induced hRSV-neutralizing antibodies in macaques, but not in mice, after repeated doses (Correia *et al*, 2014).

Because the recently solved structure of hMPV F in the postfusion conformation revealed a high level of similarity with hRSV F, particularly in antigenic site II, we hypothesized that exchanging amino acids between the two proteins at that site should allow proper folding and expression of chimeric molecules. Indeed, proteins F-415 and F-416 displayed by EM the characteristic cone shape of postfusion hMPV F, which was used as the backbone for the chimeras (Fig EV1). In addition, by ELISA the chimeras showed the antigenic characteristics of hRSV F site II (Fig 1C). In fact, F-416 showed an affinity for Mz similar to that of hRSV F although F-416 had a 10-fold reduced affinity for mAb 47F. Nevertheless, both F-415 and F-416 were able to elicit in mice very significant antibody responses against hRSV F (Fig 3). In addition, all mice immunized with three doses of F-416 were fully protected against a hRSV challenge. Although only two mice inoculated with F-415 were protected against hRSV, those mice were the ones with higher neutralizing titers.

While the protection afforded by the chimeric proteins in the experiment of Fig 4B cannot be ascribed exclusively to antibodies, the relevance of antibodies in protection was substantiated by the results of the adoptive transfer experiment of Fig EV3. Since other studies have already demonstrated the capacity of postfusion hMPV F to induce protective immune responses in cotton rats (Cseke *et al*, 2007), hamsters (Herfst *et al*, 2007), mice (Aerts *et al*, 2015), and macaques (Herfst *et al*, 2008), the chimeric proteins could have the advantage over the epitope scaffolding approach in that they would confer protection simultaneously against more than one virus. This possibility however needs to be tested directly in perhaps more

permissive animal models such as cotton rats where protection in the upper and lower respiratory tract against hRSV and hMPV could be assessed. Additionally, non-human primates may be a more decisive model to test an optimized chimeric F candidate.

While this work was being performed, Wen *et al* (2016) reported a single chimeric hMPV F protein (RPM-1) that incorporated certain site II amino acids from hRSV F. In contrast to the results reported here, two high doses of RPM-1 failed to induce hRSV-neutralizing antibodies in mice and only one out of five mice immunized with RPM-1 elicited antibodies that bound faintly to hRSV F. Whether differences in the amino acids substituted in the F proteins or the different immunization protocols account for the discrepant results reported by us and by Wen *et al* (2016) requires further investigation.

Since the majority of neutralizing antibodies against hRSV in humans recognize epitopes specific to prefusion F (Magro *et al*, 2012; Ngwuta *et al*, 2015; Gilman *et al*, 2016), it was important to the test the feasibility of this conformation for grafting hMPV antigenic sites. Indeed, F-410 and F-411 showed characteristics of prefusion hRSV F by EM (Fig EV4) and by reactivity with conformation-specific mAbs (Fig EV5). At the same time, F-410 and F-411 gained reactivity, at least partially, with mAbs specific to hMPV site IV. Both proteins elicited in mice polyclonal antibody responses that bound hMPV F and neutralized this virus.

The chimeric protein approach described here undoubtedly requires further optimization. Grafting of other antigenic sites, like those specific to prefusion hRSV F, deserves being tested. The yield of certain chimeric proteins, as shown in Fig EV1, turned out to be extremely low. In that sense, it is worth noting that grafting of hRSV site II yielded proteins that could be expressed and purified while incorporation of hMPV F site II into either postfusion or prefusion hRSV F ablated the expression of the matching chimeric proteins (Fig EV1). Thus, fine tuning of changes introduced in chimeric proteins may be critical for their expression and immunogenicity. The use of other expression systems or incorporation of chimeric proteins into viral vectors (Liang *et al*, 2017) or virus particles (McGinnes *et al*, 2015) are alternatives to improve production and/ or immunogenicity of chimeric F proteins. Nonetheless, the results presented here represent a proof of principle to design chimeric proteins that may be used as single immunogens to induce cross-neutralizing and cross-protecting immune responses against the important *Pneumoviridae* members, hRSV and hMPV.

## Materials and Methods

### Cloning, expression, and purification of soluble forms of hRSV F and hMPV F and chimeric derivatives

The following pRB21 (Blasco & Moss, 1995) plasmids encoding soluble forms of either hRSV F or hMPV F proteins were used in this study:

(i)   pRB21/prefusion hRSV F (Pre-hRSV F): This plasmid encodes the ectodomain of hRSV F (residues 1–524) from the Long strain, stabilized in its prefusion conformation by incorporating the mutations described by McLellan *et al* (McLellan *et al*, 2013a) for the DS-Cav1 protein, which included addition of both an intraprotomer disulfide bridge (S155C-S290C) and two cavity-filling substitutions (S190F and V207L) (Palomo *et al*, 2016).

(ii)   pRB21/postfusion hRSV F (Post-hRSV F): This plasmid encodes residues 1–524 of the hRSV F ectodomain (Long strain) with a deletion of the fusion peptide (residues 137–146) to avoid aggregation (McLellan *et al*, 2011b; Palomo *et al*, 2016).

(iii)   pRB21/postfusion hMPV F (Post-hMPV F). This plasmid encodes residues 1–489 of the hMPV F ectodomain (NL/1/00) in which the natural cleavage site was substituted by a polybasic furin site and the first nine amino acids of the fusion peptide (residues 103–111) were deleted (Mas *et al*, 2016).

Plasmids encoding the chimeras of Fig EV1 were obtained by mutagenesis of the corresponding pRB21 plasmids using the Phusion Site-Directed Mutagenesis kit (Thermo Fisher Scientific) and designed primers, as recommended by the manufacturer. Sequences of those primers can be obtained from the authors upon request. In all cases, the foldon trimerization domain (Meier *et al*, 2004) was added at the C-terminus of the F protein ectodomain, flanked upstream by a TEV protease site and downstream by a Factor Xa protease site and 6xHis-tag.

The different plasmids were tested for transient expression of the wild-type or chimeric proteins as follows: monolayers of CV-1 cells growing in DMEM supplemented with 10% fetal calf serum (DMEM10) were infected with the furin-expressing vaccinia virus VV:bfur (Vey *et al*, 1994) (m.o.i. 10 pfu/cell) followed by transfection with 5 μg/ml of each plasmid mixed with 15 μg/ml of Lipofectamine 2000 (Invitrogen). Forty-eight hours later, the supernatants were tested for the presence of wild-type or chimeric proteins by ELISA, with the antibodies indicated in the figure legends.

For large-scale production, vaccinia virus recombinants encoding the chimeric proteins were obtained by the method of Blasco and Moss (1995) based on recombination of vRB12 with pRB21 plasmids. The vaccinia strain vRB12 lacks most of the gene encoding vp37 and hence is deficient in plaque formation under standard conditions. pRB21-derived plasmids carry a complete copy of the vp37 gene and thus, via homologous recombination with vRB12, restore vp37 function.

Proteins were produced in CV-1 cells infected with the matching vaccinia virus and purified from culture supernatants using $Ni^{2+}$ columns followed by gel filtration on a HiLoad 16/600 Superdex 200 pg column, as described (Mas *et al*, 2016).

### mAbs and Fab fragments

mAbs specific for either hRSV F or hMPV F are indicated and referenced in the figure legends. Motavizumab [a more potent derivative of palivizumab (Robbie *et al*, 2013)] was produced in HEK293 cells transfected with plasmid DNA encoding the antibody heavy and light chains and purified using Protein A-Sepharose. The Fab fragment was generated by papain digestion (mass ratio 100:1) for 4 h at 37°C, followed by passage over Protein A-Sepharose to remove the Fc fragment. Antibody 47F (Garcia-Barreno *et al*, 1989) was produced by growing the hybridoma cells in RPMI medium supplemented with 10% fetal calf serum. The antibody was purified from culture supernatants as above and its Fab fragment produced as described above.

## Surface plasmon resonance

All assays were run in a Biacore X100 instrument. An anti-6xHis mAb was covalently coupled to both the sample and reference cells of a CM5 chip at 10,000 response units (RU). Approximately 140 RU of the proteins indicated in the figure legends were bound to the anti-6xHis mAb, and then, the Fab fragments at 8–10 different protein concentrations (0.5–500 nM) were injected. Binding data were fit to a 1:1 Langmuir binding model for calculation of the kinetic parameters.

## Mouse immunization and virus challenge

Animal studies were performed under the regulations of the Spanish and European legislation concerning vivisection and the use of genetically modified organisms. Protocols were approved by the "Comité de Ética de la Investigación y del Bienestar Animal" of "Instituto de Salud Carlos III" (CBA PA 19_2012). Specific pathogen-free 6- to 8-week-old female BALB/c mice were obtained from ENVIGO (RMS, Spain). Animals were group-housed in ventilated racks under a 12-h light/12-h dark schedule at an ambient temperature of 21°C with food and water available *ad libitum*. Group size was estimated using previous published data (Mas *et al*, 2016), and no randomization was used in grouping. No blinding procedure was used. In all experiments, baseline antibodies (IgG) against both hRSV and hMPV postfusion were below the detection limit groups of 5–8 mice were inoculated i.m. bilaterally in the quadriceps muscles (50 µl/site) with the indicated amounts of proteins (see figure legends) in PBS mixed with an equal volume of CpG (Magic Mouse adjuvant, Creative Diagnostics). The number of doses in 3–4 week intervals is indicated in each figure legend. 1 or 2 weeks after the last dose, mice were bled from the submandibular vein to obtain serum samples.

For challenge studies, mice were inoculated i.n. with $4 \times 10^6$ pfu in 50 µl of PBS of the A2 strain of hRSV, which had been purified from the supernatant of hRSV-infected cells by ultracentrifugation (230,000 *g*/90 min) in a discontinuous sucrose gradient (30–45% sucrose). Five days later, mice were euthanized, lungs removed and extracts made in 5 ml of Dulbecco's medium with 2.5% fetal calf serum. hRSV titers in the lung extracts were determined by plaque assay on HEp-2 cells (Gonzalez-Sanz *et al*, 2016).

## Enzyme-linked immunosorbent assay (ELISA)

For testing mAb reactivity, 400 ng of purified mAbs in PBS was used to coat 96-well microtiter plates. Non-specific binding was blocked with 1% bovine serum albumin in PBS. Then, serial dilutions of the different proteins were added to the wells for 1 h at room temperature, followed by an excess of an anti-His mAb (Bio-Rad, MCA 1396B), streptavidin–peroxidase, and substrate (OPD, GE Healthcare). Extensive washing with water was done after each step. Optical density was read at 492 nm.

For testing mouse serum reactivity, the microtiter plates were coated with 80 ng/well of purified protein in 50 mM $Na_2HPO_4$, pH 8.0, 300 mM NaCl, and non-specific binding was blocked as above. Then, serial dilutions of sera in blocking solution were added, and bound antibodies were detected with peroxidase-labeled goat anti-mouse Ig and OPD.

When sera were tested for binding to postfusion F proteins, the foldon motif was removed from the ELISA antigen by pretreatment

### The paper explained

**Problem**

hRSV and, to a lesser extent, hMPV are the main causes of severe ALRI worldwide in very young children. Both viruses are members of the same family (*Pneumoviridae*) that share many biological and clinical characteristics, but they also have important differences. Although much needed, no vaccine is available against either virus. The F glycoprotein is the main target of neutralizing antibodies that correlate with protection; however, no significant cross-neutralization between hRSV and hMPV is observed in polyclonal responses raised against their respective F antigens.

**Results**

Based on available structures of hRSV and hMPV F, a structure-based approach was used to design chimeric F proteins in which particular antigenic sites recognized by neutralizing antibodies were swapped between the F glycoproteins of the two viruses. When inoculated into mice, the chimeric proteins induced polyclonal antibody responses that cross-neutralized both hRSV and hMPV and even protected against challenge with the virus used for grafting of the heterologous antigenic site.

**Impact**

These results demonstrate the feasibility of designing a single immunogen capable of inducing polyclonal antibody responses that neutralize both hRSV and hMPV, setting the stage for development of cross-protective vaccines against human *Pneumoviridae*.

with TEV protease and removal with $Ni^{2+}$ columns to avoid binding of anti-foldon antibodies present in the sera (Mas *et al*, 2016). In the case of sera tested for binding to prefusion hRSV F, the anti-foldon antibodies were first depleted by incubation with $Ni^{2+}$ beads coated with a MERS-CoV S1 protein containing a C-terminal foldon domain and 8xHis-tag (Pallesen *et al*, 2017).

## Microneutralization assay

GFP-expressing recombinant viruses derived from the A2 strain of hRSV (a kind gift of Mark Peeples, Columbus, OH, USA) or the NL/1/00, sublineage A1, of hMPV (a kind gift of Bernadette van den Hoogen and Ron Fouchier, Rotterdam, the Netherlands) were mixed with serial dilutions of mouse serum before being added to cultures of Vero-118 cells growing in flat-bottom microtiter plates. Twenty-four to forty-eight hours later, the medium was replaced by PBS and the GFP fluorescence measured in a Tecan microplate reader M200. Values were expressed as percent of a virus control without antibody.

## Electron microscopy (EM)

Purified proteins were applied to glow-discharged carbon-coated grids and negatively stained with 1% uranyl formate, as described (Mas *et al*, 2016). Images were recorded on a Gatan ES1000W CCD camera in a JEOL JEM-1011 microscope operated at 100 kV.

## Statistical analysis

Animal sample sizes were chosen based on literature documentation and previous results obtained with similar experiments (Mas

*et al*, 2016). The number of animals used in each study is indicated in the figure legends. No inclusion or exclusion criteria were used, and studies were not blinded to investigators or formally randomized. Results are expressed as individual values and their mean. Statistical significance was calculated by one-way ANOVA, followed by Tukey's test using GraphPad Prism 6.

**Expanded View** for this article is available online.

## Acknowledgements

We would like to thank the personnel of the Genomics and the Animal House facilities at Instituto de Salud Carlos III for their help in part of the work here presented. This work was supported in part by grants SAF2015-67033-R (JAM), PI15CIII/00024 (IM) from Instituto de Salud Carlos III, BFU 2013-43149-R (DL) from Plan Nacional I+D+I,c, and P20GM113132 (JSM) from the National Institute of General Medical Sciences of the National Institutes of Health.

## Author contributions

EO, JSM, JAM, and VM designed the study and interpreted results. EO, OC, and VM performed the experiments. IM provided the virus and supervised the challenge experiments. DL and MCT performed the electron microscopy. JAM and VM supervised the study. JSM and JAM wrote the manuscript. All authors contributed to the final version of the manuscript.

## Conflict of interest

The authors declare that they have no conflict of interest.

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
