## [Review Process File · EMBO Molecular Medicine]

Chimeric *Pneumoviridae* fusion proteins as immunogens to induce cross-neutralizing antibody responses

Eduardo Olmedillas, Olga Cano, Isidoro Martínez, Daniel Luque, María C. Terrón, Jason S. McLellan, José A. Melero and Vicente Mas

Corresponding author: José Melero, Instituto de Salud Carlos III

Review timeline:

Submission date:	26 May 2017
Editorial Decision:	07 June 2017
Rebuttal:	07 June 2017
Editorial Decision:	14 August 2017
Revision received:	05 October 2017
Editorial Decision:	23 October 2017
Revision received:	30 October 2017
Accepted:	08 November 2017

Transaction Report:

Editor: Céline Carret

1st Editorial Decision

07 June 2017

Thank you for the submission of your manuscript "Chimeric fusion proteins set the stage for human pan-Pneumoviridae vaccines". We have been unusually busy and traveling lately, which has led to unfortunate delays. I have now had the opportunity to carefully read your paper and the related literature and I have also discussed it with my colleagues and an Editorial Advisory Board member. I am afraid that we decided that the manuscript is not well suited for publication in EMBO Molecular Medicine and have therefore decided not to proceed with peer review.

While the idea to make a vaccine that covers RSV and human metapneumovirus is interesting and novel, we are not convinced at this stage that the data would be clinically significant enough for the paper to be further considered. This opinion was shared by the Editorial Advisory Board member we consulted with, who stated "Technically the manuscript seems to be well done. The problem seems to me that the results are not [meaningful] enough to be exciting: the neutralizing titers against metapneumovirus are quite low compared to the homologous protein." As this expert knows our journal well, I am afraid that we decided to return the manuscript to you at this stage.

I am sorry that I could not bring better news.

Appeal

07 June 2017

Thank you very much for your email and comments about our manuscript. We understand that a Journal like EMBO Molecular Medicine receives many more papers than those that could publish and that therefore relatively high quality and significance standards should be reached by accepted manuscripts. Nevertheless, we feel that in agreement with your comments our manuscript offers novel and interesting insights into the field of pneumovirus vaccines, particularly in view of the negative results reported by Wen et al. (PLoS One, doi: 10.1371/journal.pone.0155917). Their results seemed to rule out the possibility of eliciting cross-neutralizing antibody responses with chimeric F proteins, something clearly revoked by our results.

We agree that at present our results are still far from a clinical application. Nevertheless, as mentioned in the Discussion of our manuscript, optimization of the chimeric approach would likely provide other vaccine candidates with improved characteristics. Still, the results obtained with the chimeric metapneumovirus F protein incorporating antigenic site II from RSV F (Fig. 4) already demonstrate the feasibility of affording protection against a RSV challenge with chimeric proteins without reaching neutralizing titers as high as with the homologous protein. The metapneumovirus neutralization titers of Fig. 6 (of concern to the Editorial Board member that you contacted) are not lower than the hRSV neutralization titres found in some mouse sera of Fig. 4 and these mice were protected against the heterologous challenge. Therefore, our results already provide evidence of inducing cross-neutralizing and protecting responses with chimeric F proteins although there is still room for improvement; e.g, by grafting more than one antigenic site to the chimeras.

We therefore would like to ask if it is still possible to request the opinion of further experts before a final decision is reached about publication of our manuscript in EMBO Molecular Medicine.

2nd Editorial Decision

14 August 2017

Thank you for your patience while we proceeded with peer-review of your article. We have now heard back from two referees whom we asked to evaluate your manuscript. I apologise for the long delay due to getting appropriate referees keeping to deadline, and the summer holidays season.

You will see from the set of comments pasted below that both referees appreciated the findings and we would like to invite a major revision of your work on the basis that 1) you can perform better appropriate *in vivo* analysis (and ref.1 points to using cotton rats as a better model), and 2) tune down some over-interpretation of the data and overall significance for viral infection.

I look forward to receiving your revised manuscript.

***** Reviewer's comments *****

Referee #1 (Comments on Novelty/Model System):

For a more complete assessment of the efficacy of the chimera proteins, the authors should test the induction of neutralizing antibody and protection from RSV and MPV challenges in cotton rats. Cotton rats are permissive to both viruses while mice are only semi permissive for RSV and not permissive for MPV. It is only in cotton rats that the protection from MPV challenge can be assessed after immunization with the chimera proteins.

Referee #1 (Remarks):

This manuscript describes a potential method to produce a pneumovirus F protein vaccine candidate that will induce protective responses to both respiratory syncytial virus (RSV) and metapneumovirus (MPV). The authors have taken advantage of the availability of the crystal structures of the pre-F and post-F forms of RSV F protein and the newly reported post-F form of the MPV F protein. The approach is to make a chimera protein with amino acids specific to an F protein epitope from one virus grafted into the backbone of the other virus F protein with the goal of generating a protein containing neutralizing antibody binding sites from both viruses. The manuscript is clearly written, and the experiments are well done with appropriate controls and statistical analyses.

The study was somewhat hampered by the lack of a structure of the MPV pre-fusion F protein. Thus the authors chose to focus on grafting site II or site IV amino acids from one virus onto the backbone of the other virus. The combinations of changes made are logical. However, a major problem is that many of the chimera proteins made were very poorly expressed. Indeed, the proteins with the most logical combination of backbones and mutational changes failed to express at levels that were workable. This finding points to a major problem that may be encountered in this approach, ability of chimera proteins to fold properly and thus be expressed.

However, using the four chimera proteins that were expressed, the authors presented results that suggested the feasibility of the approach with the right combination of backbones and mutational changes. However, the authors did not clearly identify a potential vaccine candidate. Chimera F-416 is the most promising in terms of inducing potentially protective responses to the two viruses. Unfortunately this chimera is very poorly expressed (10% wild type) making its development as a vaccine problematic.

Specific points

1. The authors should discuss more fully the issue of expression levels of chimera proteins in their approach and their results.
2. The manuscript would be strengthened by a more comprehensive study of immune responses to chimera proteins, particularly F-416. For example, what was the level of protection from RSV challenge in the upper respiratory tract of mice after immunization? What levels of neutralizing antibody titers and protection from RSV or MPV challenge can be induced in cotton rats? Use of cotton rats would strengthen the study since the cotton rat is permissive to replication of both viruses in the lung and respiratory tract, in contrast to mice.
3. Figure 6, panel B: peptides are mislabelled.
4. Line 223: Shouldn't "eight" be "five" since 3 amino acids are in common?
5. Line 225: Shouldn't "RSV-like" be "MPV-like"?

Referee #2 (Remarks):

This is a straightforward, elegantly conducted and well-presented study in which, based on recently determined crystal structures of hRSV F and hMPV F, chimeric F proteins were designed in which residues of antigenic sites were swapped between the two antigens. Epitope scaffolding was shown to result in protection simultaneously against more than one virus:

- Antigenic changes in the scaffolding chimeric F proteins were studied by ELISA with monoclonal antibodies.
- Mouse immunization with chimeric proteins induced cross-neutralizing serum antibody.
- Mouse immunization with these chimeric proteins also resulted in protection against challenge with the virus from which the heterologous antigenic site was grafted.

This is a proof-of-principle study that shows that it is possible to induce family-wide cross protection with chimeric proteins in which antigenic sites have been swapped between protection inducing proteins from different viruses within the Pneumoviridae family.

Although polyclonal sera after infection with one pneumovirus do not neutralize the other, it is correctly pointed out that previously cross-reactive virus neutralizing monoclonal antibodies have been identified.

It is claimed that these results demonstrate the feasibility of 'universal vaccines' that could protect against infections by all human Pneumoviridae. As to date - unlike the situation for influenza viruses and HIV- no more than two (important) members of this family have been identified in humans (all be it with two subtypes each) which are antigenically rather stable over time, and 'the chimeric protein approach described here undoubtedly requires further optimization', this claim may be considered somewhat audacious at this stage. Toning down the title and changing the last paragraph of the discussion section into: 'results presented here demonstrate the feasibility of using chimeric proteins to design single immunogens that could induce cross neutralizing and cross-protecting

immune responses against the important human Pneumoviridae members hRSV and hMPV', would seem more appropriate.

Major point

The mechanism of protective immunity which is observed in the immunized mice, is ascribed to virus neutralizing antibodies induced by the chimeric proteins. This is however not shown formally. It could theoretically be due to another immune mechanism, like T cell mediated immunity. To formally proof that the protection induced is antibody mediated, adoptive transfer experiments with post-immunization serum antibodies (and T cells) should have been carried out. Either additional adoptive transfer experiments should be carried out, or an adequate statement should be made in the discussion section to address this point.

Minor point

The lack of protection and enhanced disease after FI RSV vaccination of children can no longer (solely) be attributed to low- or non-neutralizing antibody.

1st Revision - authors' response

05 October 2017

Reviewer #1

This manuscript describes a potential method to produce a pneumovirus F protein vaccine candidate that will induce protective responses to both respiratory syncytial virus (RSV) and metapneumovirus (MPV). The authors have taken advantage of the availability of the crystal structures of the pre-F and post-F forms of RSV F protein and the newly reported post-F form of the MPV F protein. The approach is to make a chimera protein with amino acids specific to an F protein epitope from one virus grafted into the backbone of the other virus F protein with the goal of generating a protein containing neutralizing antibody binding sites from both viruses. The manuscript is clearly written, and the experiments are well done with appropriate controls and statistical analyses.

The study was somewhat hampered by the lack of a structure of the MPV pre-fusion F protein. Thus the authors chose to focus on grafting site II or site IV amino acids from one virus onto the backbone of the other virus. The combinations of changes made are logical. However, a major problem is that many of the chimera proteins made were very poorly expressed. Indeed, the proteins with the most logical combination of backbones and mutational changes failed to express at levels that were workable. This finding points to a major problem that may be encountered in this approach, ability of chimera proteins to fold properly and thus be expressed.

However, using the four chimera proteins that were expressed, the authors presented results that suggested the feasibility of the approach with the right combination of backbones and mutational changes. However, the authors did not clearly identify a potential vaccine candidate. Chimera F-416 is the most promising in terms of inducing potentially protective responses to the two viruses. Unfortunately this chimera is very poorly expressed (10% wild type) making its development as a vaccine problematic.

Specific points

1. The authors should discuss more fully the issue of expression levels of chimera proteins in their approach and their results.

Response: Our initial objective was to report the induction of cross-neutralizing antibody responses as a proof-of-principle for the use of single immunogens to protect against both hRSV and hMPV. We agree that our approach for the design and production of chimeric F proteins requires further improvements. An extended comment about this point has been added in the new version of the manuscript (line 281-293), suggesting several alternatives.

2. The manuscript would be strengthened by a more comprehensive study of immune responses to chimera proteins, particularly F-416. For example, what was the level of protection from RSV challenge in the upper respiratory tract of mice after immunization? What levels of neutralizing antibody titers and protection from RSV or MPV challenge can be induced in cotton rats? Use of cotton rats would strengthen the study since the cotton rat is permissive to replication of both viruses in the lung and respiratory tract, in contrast to mice.

Response: We appreciate the use of cotton rats as a more permissive animal model for hRSV, although neither mice nor cotton rats fully reproduce the pathology of hRSV or hMPV infections in humans. At this moment the logistic to use cotton rats in our animal facilities is totally impossible. It is our intention to carry out a more extensive protection study in cotton rats and perhaps in other animal model system once a suitable vaccine candidate is identified.

3. Figure 6, panel B: peptides are mislabeled.

Response: Thank you for pointing out this error. It has now been corrected.

4. Line 223: Shouldn't "eight" be "five" since 3 amino acids are in common?

Response: Agree. It has been changed.

5. Line 225: Shouldn't "RSV-like" be "MPV-like"?

Response: Agree and corrected.

Reviewer #2

This is a straightforward, elegantly conducted and well-presented study in which, based on recently determined crystal structures of hRSV F and hMPV F, chimeric F proteins were designed in which residues of antigenic sites were swapped between the two antigens. Epitope scaffolding was shown to result in protection simultaneously against more than one virus:

- Antigenic changes in the scaffolding chimeric F proteins were studied by ELISA with monoclonal antibodies.
- Mouse immunization with chimeric proteins induced cross-neutralizing serum antibody.
- Mouse immunization with these chimeric proteins also resulted in protection against challenge with the virus from which the heterologous antigenic site was grafted.

This is a proof-of-principle study that shows that it is possible to induce family-wide cross protection with chimeric proteins in which antigenic sites have been swapped between protection inducing proteins from different viruses within the Pneumoviridae family.

Although polyclonal sera after infection with one pneumovirus do not neutralize the other, it is correctly pointed out that previously cross-reactive virus neutralizing monoclonal antibodies have been identified.

It is claimed that these results demonstrate the feasibility of 'universal vaccines' that could protect against infections by all human Pneumoviridae. As to date - unlike the situation for influenza viruses and HIV- no more than two (important) members of this family have been identified in humans (all be it with two subtypes each) which are antigenically rather stable over time, and 'the chimeric protein approach described here undoubtedly requires further optimization', this claim may be considered somewhat audacious at this stage. Toning down the title and changing the last paragraph of the discussion section into: 'results presented here demonstrate the feasibility of using chimeric proteins to design single immunogens that could induce cross neutralizing and cross-protecting immune responses against the important human Pneumoviridae members hRSV and hMPV', would seem more appropriate.

Major point

The mechanism of protective immunity which is observed in the immunized mice, is ascribed to virus neutralizing antibodies induced by the chimeric proteins. This is however not shown formally. It could theoretically be due to another immune mechanism, like T cell mediated immunity. To formally proof that the protection induced is antibody mediated, adoptive transfer experiments with post-immunization serum antibodies (and T cells) should have been carried out. Either additional adoptive transfer experiments should be carried out, or an adequate statement should be made in the discussion section to address this point.

Response: We have performed the adoptive transfer experiment proposed by this reviewer and the results are presented in the new Fig. EV3. The new data are mentioned in a new paragraph (lines 162-174) of the amended manuscript. These results clearly demonstrate that passive transfer of sera from mice inoculated with chimeric proteins, particularly F-416, partially protected mice against a new virus challenge, even if virus reduction was not as great as with sera from mice inoculated originally with hRSV F. Therefore, although the protection afforded by the chimeric proteins in the

experiment of Fig. 4B could not be ascribed exclusively to antibodies (as rightly pointed out by the reviewer), the results of the new Fig. EV3 substantiate the relevance of antibodies for the protection observed with chimeric proteins.

Minor point

The lack of protection and enhanced disease after FI RSV vaccination of children can no longer (solely) be attributed to low- or non-neutralizing antibody.

Response: Agree. The formalin-inactivated virus vaccine experience has been omitted from the Introduction since it was only marginally related to our study.

3rd Editorial Decision

23 October 2017

Thank you for the submission of your revised manuscript to EMBO Molecular Medicine. We have now received the enclosed report from the referee who was asked to re-assess it. As you will see the reviewer is now supportive and I am pleased to inform you that we will be able to accept your manuscript pending editorial final amendments.

1) Please address the referee 1's comments. Please provide a letter INCLUDING the reviewer's report and your detailed responses to their comments (as Word file). We would appreciate it if you could have some discussion to acknowledge the limitations of not using cotton-rats in terms of vaccine protective capacities.

***** Reviewer's comments *****

Referee #1 (Comments on Novelty/Model System for Author):

The authors of this revision of a previously submitted manuscript have responded adequately to most of the suggestions made by both reviewers. However, the suggestion that the authors test their chimera proteins as vaccine candidates in a permissive animal model, cotton rats, for protection from replication of metapneumovirus was not pursued.

The response was that cotton rats do not fully reproduce the pathology of hRSV or hMPV. Perhaps, but cotton rats are permissive enough to allow replication in lungs of hMPV as well as hRSV. They are widely used as animal models for both viruses for assays of pathology and virus replication in the lungs. As it stands, the manuscript only tests one set of chimera proteins, chimeras based on MPV F, in mice (semi-permissive only for hRSV) for effects on RSV replication in the lung. Thus the study could be viewed as incomplete as the chimera proteins based on RSV F with MPV sequences were not tested for protection from MPV replication.

Also the argument was made that the lab was not set up for cotton rat studies. Indeed, there are very few labs that can do these studies but most contract the project to companies that have the capability to do these studies or they establish collaborations with labs that can do them.

Line 277: shouldn't "hMPV" be hRSV since the protein is pre-fusion RSV?

Referee #1 (Remarks for Author):

I view the study as incomplete for reasons stated in the comments to the authors. In addition, testing for protection from RSV replication in mice is not a good test of protection since mice are only semi-permissive to RSV. However, the editors may consider the manuscript acceptable as a test of feasibility. In addition, extending the study to cotton rats may be prohibitively expensive.

Referee #1

The authors of this revision of a previously submitted manuscript have responded adequately to most of the suggestions made by both reviewers. However, the suggestion that the authors test their chimera proteins as vaccine candidates in a permissive animal model, cotton rats, for protection from replication of metapneumovirus was not pursued.

The response was that cotton rats do not fully reproduce the pathology of hRSV or hMPV. Perhaps, but cotton rats are permissive enough to allow replication in lungs of hMPV as well as hRSV. They are widely used as animal models for both viruses for assays of pathology and virus replication in the lungs. As it stands, the manuscript only tests one set of chimera proteins, chimeras based on MPV F, in mice (semi-permissive only for hRSV) for effects on RSV replication in the lung. Thus the study could be viewed as incomplete as the chimera proteins based on RSV F with MPV sequences were not tested for protection from MPV replication.

Also the argument was made that the lab was not set up for cotton rat studies. Indeed, there are very few labs that can do these studies but most contract the project to companies that have the capability to do these studies or they establish collaborations with labs that can do them.

***Response:** We overlooked in our previous responses the referee's suggestion to use cotton rats as a model system for hMPV infection. Indeed, cotton rats are more permissive for both hRSV and hMPV and therefore are a better model than mice to evaluate protection against both viruses. The reason for not using cotton rats in the present study was an issue of availability and logistic in our animal house (in addition to expenses mentioned below), as pointed out in our initial response. To underlie this caveat of our current study, new sentences have been added in lines 266-269 (highlighted in yellow) indicating that optimized chimeric F proteins should be tested in cotton rats, and perhaps in non-human primates, for protection of the upper and lower respiratory tract against hRSV and hMPV challenge.*

Line 277: shouldn't "hMPV" be hRSV since the protein is pre-fusion RSV?

***Response:** Thank you for pointing out this error. It has now been corrected (line 280, highlighted in yellow). Also two missing words in line 273 have been added (highlighted in yellow)*

Referee #1 (Remarks for Author):

I view the study as incomplete for reasons stated in the comments to the authors. In addition, testing for protection from RSV replication in mice is not a good test of protection since mice are only semi-permissive to RSV. However, the editors may consider the manuscript acceptable as a test of feasibility. In addition, extending the study to cotton rats may be prohibitively expensive.

***Response:** As indicated before, we acknowledge the need to substantiate further the protection afforded by chimeric F proteins in animal models other than mice. However, as noted by the reviewer, the added costs restrict those studies to a handful of optimized vaccine candidates as we intend to follow in the future.*

YOU MUST COMPLETE ALL CELLS WITH A PINK BACKGROUND ↓
PLEASE NOTE THAT THIS CHECKLIST WILL BE PUBLISHED ALONGSIDE YOUR PAPER

Corresponding Author Name: José A. Melero
Journal Submitted to: EMBO Molecular Medicine
Manuscript Number: EMM-2017-08078-V4